# Association of Paternity Leave with Impaired Father–Infant Bonding: Findings from a Nationwide Online Survey in Japan

**DOI:** 10.3390/ijerph19074251

**Published:** 2022-04-02

**Authors:** Shuhei Terada, Takeo Fujiwara, Erika Obikane, Takahiro Tabuchi

**Affiliations:** 1Department of Global Health Promotion, Tokyo Medical and Dental University, Tokyo 113-8519, Japan; terada.hlth@tmd.ac.jp; 2Department of Mental Health, Graduate School of Medicine, The University of Tokyo, Tokyo 113-0033, Japan; obikane-e@ncchd.go.jp; 3Cancer Control Center, Osaka International Cancer Institute, Osaka 541-8567, Japan; tabuti-ta@mc.pref.osaka.jp

**Keywords:** bonding impairment, childcare, father–infant bonding, Japan, paternity leave, psychological distress

## Abstract

Although the number of fathers taking childcare leave is increasing, the impact of paternity leave on father–infant bonding remains to be fully investigated. This study aimed to assess the association between paternity leave and father–infant bonding among fathers with children under two years old. A cross-sectional study was performed using data from the Japan COVID-19 and Society Internet Survey study, a nationwide online survey conducted between July and August 2021 (N = 1194). Father–infant bonding was assessed by the Japanese version of the Mother–Infant Bonding Scale (MIBS-J), which comprised two subscales (lack of affection (LA), and anger and rejection (AR)), with a higher score denoting poor bonding. Four hundred (33.5%) fathers self-reported taking childcare leave. Paternity leave was associated with higher total MIBS-J score and AR score after adjusting for covariates (coefficient 0.51; 95% confidence interval (CI) 0.06–0.96, coefficient 0.26; 95% CI 0.03–0.49, respectively), but not with LA score (coefficient 0.10; 95% CI −0.13–0.34). There was no trend in the association between paternity leave and total MIBS-J score by children’s age group (*p* for trend = 0.98). Paternity leave was associated with impaired bonding, especially with increased anger and rejection, among fathers with children under two years of age.

## 1. Introduction

Bonding, the emotional tie with or love of parents toward their child, plays a central role in parenting [1,2]. This concept was first theorized in the mother–child relationship and was later extended to the father–child relationship [3]. Previous research on this subject has found that successful father–infant bonding was associated with better infant growth and development, including reduced cognitive delays, increased weight gain in preterm infants, and higher breastfeeding rates [3,4,5]. Furthermore, the establishment of father–infant bonding may encourage fathers to be more actively involved in child rearing [3], leading to higher academic achievement, better socioeconomic status, and fewer behavioral problems among children [5,6]. Thus, father–infant bonding is important for the psychosocial well-being of the child throughout his or her life. However, factors that promote or deteriorate father–infant bonding have not been adequately studied so far [3].

Paternity leave, a father’s temporary absence from work to take care of his newborn child, has been suggested to increase fathers’ involvement with childcare [7]. Currently, most Organisation for Economic Co-operation and Development (OECD) member countries have launched a legal paternity leave policy to promote gender equality in childcare [8]. Japan is no exception; fathers with infants have the right to take childcare leave until the child reaches the age of 1 year (until the day before the child’s first birthday). The proportion of fathers taking paternity leave has been increasing in Japan, from 0.33% in 2002 to 12.7% in 2019 [9], with the revision of the Child Care and Family Care Leave Act in 2010 and then in 2017 that allows more flexible paternity leave. While paternity leave-taking has been shown to encourage higher levels of father’s engagement and responsibility [7], the impact of paternity leave on father–infant bonding is not fully understood.

Paternity leave may affect subsequent father–infant bonding by providing fathers with time to be involved with their children [10]. A cohort study in US revealed that paternal leave taking before the child turn one year old, especially one with a duration of two weeks or more, was positively correlated with children’s perceptions of father’s involvement, father–child closeness and father–child communication at age 9 [10]. In contrast, a cohort study in Germany reported the negative association of paternity leave and father–infant bonding at 14 months postpartum [11]. In this previous study, time spent on childcare did not mediate the association of paternity leave and father–infant bonding, indicating that the underlying mechanism may be more complex [11]. Moreover, these studies were limited by insufficient information on confounders such as postnatal depression, fathers’ perceived social support by their own parents, or workplace social capital, which have been associated with lower father–infant bonding in previous studies [12,13,14,15].

In Japan, while the use of paternity leave is spreading rapidly, fathers on childcare leave may not have enough parenting skills, which may induce lack of confidence in parenting [16]. According to a survey conducted in 2014 on Japanese fathers with children under the age of 6 years old, 44% said they were not confident in how to interact with their children [17]. Another cross-sectional study in New Zealand found that about 30 to 40% of fathers expressed moderate or low levels of confidence in dealing with children’s behavior and emotions [18]. Fathers in Finland also reported lower parenting self-efficacy than mothers [19]. Considering the evidence that parenting self-efficacy and self-confidence affects bonding [20], we hypothesized that paternity leave-taking may have negative impact on father–infant bonding in Japan.

Based on the gap in the literature and our hypothesis, the aim of this study was to examine the association of paternity leave taking and bonding between father and children under two years of age using a nationwide survey data in Japan.

## 2. Materials and Methods

### 2.1. Study Design and Sample

This study is part of the Japan COVID-19 and Society Internet Survey (JACSIS), a nationwide, cross-sectional, online survey [21]. Briefly, the JACSIS survey consisted of three surveys in the following target populations: (1) youth and adults aged 15 to 79, (2) women who are currently pregnant or postpartum, and (3) adults living in single-parent households [22], all of which were conducted in 2021. For each survey, the study samples were obtained from the pooled panels of an internet research company (Rakuten Insight, Inc., Hong Kong, which holds approximately 2.3 million panelists who voluntarily registered for small incentive points they receive when completing questionnaires). We used data from the pregnancy and maternity survey, which also includes data on male participants whose partners were currently pregnant or postpartum. After screening 440,323 panelists on 24 July 2021, the research agency identified 14,086 women and 3346 men as eligible who (or whose partner) had given birth after July 2019 or expected to give birth by December 2021. The survey ended with a total of 10,000 respondents; 8047 (response rate: 57.1%) women and 1953 (response rate: 56.8%) men responded between 28 July to 30 August 2021. We restricted our sample to all fathers whose youngest child was under 2 years, excluding those whose youngest child was not yet born (*n* = 560) or over 2 years old (*n* = 56). We also excluded 143 fathers with invalid responses defined as follows: respondents who did not select a particular option (out of five) when asked to do so in a dummy question (*n* = 123); those who reported abusing all seven substances (alcohol, sleeping medications, opioids, sniffing patient thinner, legal-high drugs, marijuana, and cocaine/heroine) (*n* = 5); or those who selected all 16 past medical history in the list (*n* = 20). Following these exclusions, data from 1194 respondents were included in the analysis (Figure 1).

### 2.2. Variable Measurement

#### 2.2.1. Father–Infant Bonding

Father–infant bonding was assessed by the Japanese version of Mother–infant Bonding Scale (MIBS-J) [23]. It was developed based on Kumar’s mother–infant bonding questionnaire [24] and validated in a previous study [25]. It consists of 10 questions, each of which is scored from 0 to 3 points, for a total score of 0 to 30 points. Scores for some items are reversed (items 2, 3, 4, 5, 7, and 9), with higher score indicating poorer father–infant bonding. MIBS-J has been shown to have a two-factor structure, “lack of affection (LA)” (items 1, 6, 8, and 10) and “anger and rejection (AR)” (item 2, 3, 5, and 7) [23]. Each subscale indicates a different aspect of father’s mental health [25]. Therefore, both total score and two subscales were analyzed in the study. Although MIBS-J was originally developed for mothers of infants between 1 and 4 months of age, cross-sectional study in Japan showed the same MIBS factor structure in a sample of parents of older children (mean age 3.3 years (standard deviation 2.7 years)) and that it could be used with validity both mothers and fathers and with children of a wide age range [26].

#### 2.2.2. Paternity Leave

The usage of paternity leave was asked by the following single question: “Did you take childcare leave after your partner gave birth this time?” and the respondents chose their answer (yes or no). Fathers’ self-reported use of leave around childbirth was employed in previous studies [10,27,28].

#### 2.2.3. Covariates

As for paternal characteristics, age at survey, educational attainment (university or more, some college, high school or less, other), current job (employee, self-employed, part-time or inoccupation), and psychological distress assessed by K6 [29] were included. K6, which contains six items with total scores ranging from 0 to 24, was used as a proxy for paternal postnatal depression and anxiety [12,13,14]. We included family characteristics such as household income (<JPY5 million, JPY5-7.9 million, ≥JPY8 million; JP110 ≈ USD1), maternal jobs (inoccupation, having jobs including maternal leave), the total number of children (1, 2, 3, and ≥4), the youngest child’s age (<6 months, 6–12 months, 12–18 months, and 18–24 months), and family function. We used the Family APGAR Questionnaire [30] to assess the family function on a scale from 0 to 10, with higher scores indicating better family relationship [31]. We included fathers’ perceived social support from their own parents [15] by asking whether they receive sufficient support from grandparents (yes, no) and whether their partner used the traditional support system called “satogaeri bunben” (return to the mother’s family home for delivery and postpartum rest) (yes, no) [32].

### 2.3. Statistical Analyses

Descriptive statistics for the group taking paternity leave and the group not taking paternity leave were calculated and compared using the *t*-test for continuous variables and the chi-square test for categorical variables. To examine the association of paternity leave and father–infant bonding, simple and multiple linear regression models were estimated, yielding coefficient and 95% confidence intervals for the total MIBS-J, LA, and AR, respectively. We found no evidence of multicollinearity using variance inflation factor between included variables (all vif < 2). In addition, we performed a stratified analysis to evaluate whether the usage of paternity leave has a different impact on father–infant bonding depending on the age of the child (<6 months, 6–12 months, 12–18 months, and 18–24 months). Two-sided *p*-value of < 0.05 was considered statistically significant. All analyses were conducted using STATA MP version 16.0 (STATA Corporation, College Station, TX, USA).

### 2.4. Ethics Approval and Consent to Participants

Informed consent was obtained electronically before study participants answered the questionnaire. The study was approved by the Institutional Review Board of the Osaka International Cancer Institute (Protocol Number 20084). All methods in this study were carried out in accordance with relevant guidelines and regulations.

## 3. Results

Table 1 describes the demographic characteristics of the respondents. The prevalence of paternity leave was 33.5% (*n* = 400). The average paternal age was similar between the paternity leave-taking group (35.4 years, standard deviation [SD] = 5.3) and non-leave-taking group (35.5 years, SD = 5.2). More than 90% of all respondents were employees. Half of the respondents were fathers of their first child and 60% were fathers whose youngest child was under the age of one. The mean K6 score was 4.3 (SD = 5.4) in the group with paternity leave and 4.2 (SD = 5.3) in the group without leave. Fathers who took paternity leave tended to report that social support from grandparents was insufficient.

Table 2 reports the association of paternity leave usage with father–infant bonding. The crude model showed a significant association of paternity leave with total MIBS-J score and AR score, but not with LA score (β = 0.61; 95% confidence interval [CI]: 0.05 to 1.17, β = 0.29; 95% CI: 0.02 to 0.56, and β = 0.15; 95% CI: −0.12 to 0.42, respectively). After adjustment for paternal characteristics, family characteristics, social support from grandparents, and use of satogaeri bunben, the same trend remained (β = 0.51; 95% CI: 0.06 to 0.96, β = 0.26; 95% CI: 0.03 to 0.49, and β = 0.10; 95% CI: −0.13 to 0.34, respectively).

The results from the stratified analysis by youngest child’s age are reported in Table 3. When child’s age was stratified by 6-month increments, the mean MIBS-J score was 4.7 (<6 months), 4.1 (6–12 months), 4.6 (12–18 months), and 5.9 (18–24 months), respectively. Although the association became null, the point estimate was similar in each group (β = 0.42 (child age < 6 month), β = 0.58 (6–12 months), β = 0.48 (12–18 months), β = 0.66 (18–24 months), respectively). There was no trend in the association between paternity leave and total MIBS-J score by child’s age group (*p* for trend = 0.98).

## 4. Discussion

In this study, we found that the use of paternity leave was positively associated with father–infant bonding impairment, especially anger and rejection. This association was similar for fathers of children in any age group under two years old, suggesting that paternity leave-taking in Japan may deteriorate father–infant bonding regardless of the child’s age.

The result of this study is consistent with that of previous study which examined the association between paternity leave and father–infant bonding impairment [11]. A prospective cohort study in Germany showed that paternity leave taking (mean duration of leave: 2.4 months) was associated with father–infant bonding impairment at 14 months postpartum [11]. As occupational demand may be lower during paternity leave, and that father’s lower work demand was associated with higher MIBS-J anger score at 4 months postpartum in Japan [25], low occupational demand due to paternity leave may explain the association between paternity leave and father–infant bonding impairment.

There are three other possible mechanisms of paternity leave taking against father–infant bonding. First, a lack of confidence in parenting may contribute to bonding impairment. A cross-sectional survey in 2002 revealed that about 70% of Japanese fathers with children under 3 years old felt that their knowledge of childcare was inadequate. Additionally, nearly 50% felt that there are no role models in their parenting [16]. Another survey in 2014 reported that about half of fathers with children under 6 years old lacked confidence in the ability to raise their children [17]. By analogy with the association between maternal confidence in caregiving and mother–infant bonding [33], fathers’ lack of confidence due to insufficient knowledge and absence of role models may lead to father–infant bonding impairment during paternity leave.

Secondly, paternal loneliness during paternity leave may deteriorate father–infant bonding [34]. It was reported that only about half of fathers have a friend they can talk to about childcare and 60% of their friends are colleagues from work [17]. While a father is on leave, his friends are at work, so he may have less chance to see his friends. Furthermore, fathers taking paternity leave are still a minority in Japan, and there are few opportunities to meet and talk with other fathers on leave while at the park or childcare salons. Loneliness is closely related with the negative emotions such as sadness, disconnection, fear, worry, and anger [35]. People in the US who tweeted the words “lonely” or “alone” had more posts associated with anger than the control group (Cohen’s d = 0.95) [36]. Thus, fathers who feel lonely during paternity leave may be more prone to feelings of anger toward their children. Third, fathers taking childcare leave might feel guilty for being absent from work [37,38]. While the number of men taking paternity leave has increased with the revision of the Child Care and Family Care Leave Act, it has not immediately eradicated the deep-rooted gender division of labor [39]. According to a Japanese survey in 2014, 26.7% of male workers without paternity leave answered that leave-taking can be disruptive to the workplace [17]. In our current study, about 40% of fathers who did not take paternity leave answered that taking paternity leave would place a burden on their colleagues (data not shown). The problem is that there are generally no replacements in the workplace when men take paternity leave. This means that either their colleagues have to take over the absent employee’s work or the tasks pile up during their leave [37]. As a result, fathers on leave may suffer from guilt coupled with a fear of negative impact on promotion, which may in turn make them more prone to feelings of anger toward their children. Several social measures during the COVID-19 pandemic, such as the loss of opportunities to improve parenting skills [40], introduction of social distancing policy [41], and changes in work style and workload [42], might have worsened the association between paternity leave and father–infant bonding through the mechanisms we have discussed. Further study is needed to elucidate how paternity leave affects father–infant bonding, using a prospective design that also incorporates assessments of parenting skills (e.g., Toddler Care Questionnaire) [43], loneliness (e.g., UCLA Loneliness Scale) [44], and workplace social capital.

Several limitations of this study need to be considered when interpreting the results. First, the sample may not be representative of all populations as we recruited participants via an internet research agency. Our sample leaned toward a higher education level with full-time workers compared to that of the national census in Japan [45]. Moreover, given that vulnerable population with low socioeconomic status are more likely to lack adequate knowledge and confidence in childcare, our results might underestimate the impact of paternity leave on father–infant bonding. Second, causation is not clear due to the cross-sectional nature of the study. However, it is hard to assume that fathers with impaired bonding with infants will take more paternity leave. Third, we did not obtain information on the duration of paternity leave. According to a national survey in 2015, about 80% of all fathers taking childcare leave took only a total of less than two weeks [46]. Given the report that father–infant bonding deteriorates with a longer duration of paternity leave [11], our results in a sample in which the majority of participants are assumed to have taken similar short-term leave might underestimate the negative effect of paternity leave on father–infant bonding impairment. Finally, information on workplace social capital and characteristics of female partner (such as age or history of psychological disorders), which might be confounders, were not assessed in this study.

## 5. Conclusions

The present study found that fathers who took paternity leave were more likely to report impaired bonding, particularly in the aspects of increased emotion of anger and rejection, with children under two years of age. The current findings have possible social implications for promoting father–infant bonding in the context of the rapid increase in the use of paternity leave. From April 2022, various measures will be taken to make it easier for both parents to take childcare leave in Japan, including more flexible framework for taking childcare leave, obligation for employers to confirm the intention of employees to take childcare leave, and disclosure of the status of childcare leave taken [47]. This may lead to an increase in the number of fathers taking childcare leave in the near future. However, the experience of spending a prolonged duration of time with children during paternity leave can still be challenging and demanding for fathers who are usually dedicated to their work. Interventions that mitigate the negative impact of paternity leave would prevent deteriorating father–infant bonding. For example, encouraging fathers before taking paternity leave (such as during maternal hospitalization) to participate in parenting classes may not only help them acquire parenting skills, but also build a social network outside the workplace and prevent loneliness during paternity leave. To make it easier for fathers on leave to attend local childcare salons, it might be advisable to have “father–child only” hours or assign male childcare workers.

## Figures and Tables

**Figure 1 ijerph-19-04251-f001:**
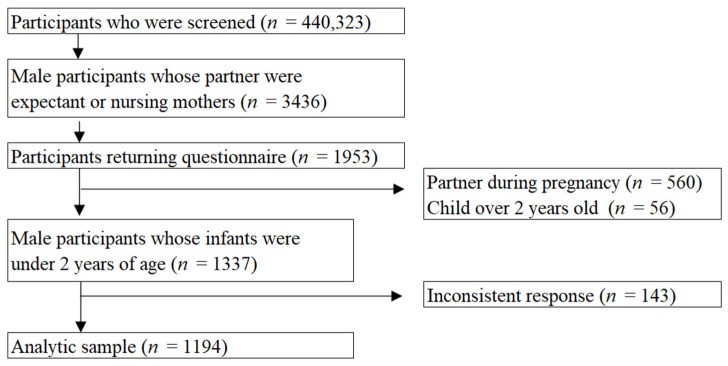
Sample flow chart.

**Table 1 ijerph-19-04251-t001:** Demographics of study participants (*n* = 1194).

		Paternity Leave (−) (*n* = 794; 66.5%)		Paternity Leave (+) (*n* = 400; 33.5%)		
		N or Mean	% or SD	N or Mean	% or SD	*p* Value ^b^
Paternal age (year)		35.5	5.2	35.4	5.3	0.80
Paternal educational attainment	University or more	607	76.5	321	80.3	
	Some college	93	11.7	43	10.8	
	High school or less	93	11.7	35	8.8	
	Other	1	0.1	1	0.3	0.37
Paternal job category	Employee	739	93.1	384	96.0	
	Self-employed	31	3.9	12	3.0	
	Part-time or inoccupation	24	3.0	4	1.0	0.06
Paternal K6		4.2	5.3	4.3	5.4	0.82
Household income (Million yen)	–4.9	115	14.5	46	11.5	
	5.0–7.9	327	41.2	160	40.0	
	8.0–	352	44.3	194	48.5	0.24
Maternal occupation	Inoccupation	309	38.9	147	36.8	
	Having jobs (including maternal leave)	485	61.1	253	63.3	0.53
Number of children	1	383	48.2	205	51.3	
	2	291	36.7	151	37.8	
	3	91	11.5	37	9.3	
	4–	29	3.7	7	1.8	0.17
Youngest child’s age	<6 months	224	28.2	115	28.8	
	<12 months	279	35.1	139	34.8	
	<18 months	211	26.6	110	27.5	
	<24 months	80	10.1	36	9.0	0.93
Family function (Family APGAR)		7.1	2.9	6.9	3.1	0.33
Social support from grandparents	No	230	29.0	168	42.0	
	Yes	564	71.0	232	58.0	<0.001 *
Satogaeri bunben ^a^	No	545	68.6	271	67.8	
	Yes	249	31.4	129	32.3	0.10

^a^ Traditional support system for perinatal women in which pregnant women return to their family home for delivery and postpartum rest. ^b^ The *p*-value was calculated using the *t*-test for continuous variables and the chi-square test for categorical variables. * indicates that the coefficient is significant at the 0.05 level (bilateral).

**Table 2 ijerph-19-04251-t002:** Association of paternity leave taking and father–infant bonding impairment.

		Crude Model				Adjusted Model			
		β	95% CI			β	95% CI		
Outcome									
Total score	Paternity leave	0.61 *	0.05	to	1.17	0.51 *	0.06	to	0.96
Lack of affection	Paternity leave	0.15	−0.12	to	0.42	0.10	−0.13	to	0.34
Anger and rejection	Paternity leave	0.29 *	0.02	to	0.56	0.26 *	0.03	to	0.49

* The coefficient is significant at the 0.05 level (bilateral). Adjusted model includes following variables: paternal age, paternal educational attainment, paternal job category, paternal K6, household income, maternal current occupation, number of children, child’s age, family function, grandparents’ support, and satogaeri bunben.

**Table 3 ijerph-19-04251-t003:** Stratified analysis of the association between paternity leave taking and father–infant bonding by infant age.

		Infant Age											
	<6 Months (*n* = 339)			6–12 Months (*n* = 418)			12–18 Months (*n* = 321)			18–24 Months (*n* = 116)			*p* for Trend
Outcome	Mean (SD)	β	95% CI	Mean (SD)	β	95% CI	Mean (SD)	β	95% CI	Mean (SD)	β	95% CI	
MIBS-J	4.7 (4.7)	0.42	−0.44 to 1.28	4.1 (4.6)	0.58	−0.22 to 1.38	4.6 (4.9)	0.48	−0.36 to 1.31	5.0 (4.3)	0.66	−0.87 to 2.20	0.98
LA	1.8 (2.3)	0.09	−0.36 to 0.55	1.5 (2.2)	0.22	−0.20 to 0.63	1.7 (2.3)	−0.03	−0.48 to 0.41	1.7 (2.0)	0.21	−0.53 to 0.94	0.33
AR	2.3 (2.2)	0.17	−0.27 to 0.61	2.0 (2.3)	0.28	−0.13 to 0.68	2.4 (2.3)	0.34	−0.09 to 0.77	2.7 (2.1)	0.27	−0.52 to 1.06	0.19

MIBS-J, the Japanese version of Mother–Infant Bonding Scale; LA, lack of affection; AR, anger and rejection; SD, standard deviation. All models were adjusted with the following variables: paternal age, paternal educational attainment, paternal job category, paternal K6, household income, maternal current occupation, number of children, family function, grandparents’ support, and satogaeri bunben.

## Data Availability

The data presented in this study are available on request from the corresponding author. The data are not publicly available due to ethical restrictions.

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
