# Peer review of "Association of Paternity Leave with Impaired Father–Infant Bonding: Findings from a Nationwide Online Survey in Japan"

_ijerph, 2022, doi:10.3390/ijerph19074251_

Round 1
Reviewer 1 Report
The article is very interesting, it raises an important issue related to the use of paternity leave by fathers and its impact on the bond with the child.
Article is well-organised. The layout of the article is typical of original papers and includes all required parts. All sections are well developed and described.
The division of the ”Results” and ”Discussion” reflect the goals of the work set by the Authors.
References list includes mostly current items (from the last 10 years) and have been prepared according to the rules for MDPI Journals.
Overall, the article is well-written and easy to understand. The Authors achieved their goals and drew the correct conclusions. The authors also indicated several limitations of the study those need to be considered when interpreting the results
In the light of the results of the study by both the Authors and those from the literature, and in the face of the increasing number of men taking paternity leave, it is correct to infer the Authors to continue and extend the scope of future research. Moreover, the implementation of education, assistance and support systems for fathers is justified.
Therefore, the results obtained by the Authors and conclusions should be published.
Author Response
We appreciate your positive comments on our study. We continue to conduct further studies related to the area.
Reviewer 2 Report
Main message of the article
The authors present a study on the impact of paternity leave on father-infant bonding in children under two years of age in the Japanese context. Adopting a modified version of the Mother-Infant Bonding Scale), the authors assessed the quality of the father-infant bonding in terms of lack of affection, anger, and rejection. Overall, results outlined that paternity leave is associated with impaired bonding, showing increased levels of anger and rejection. This work outlines critical points that require attention in the life-work balance system and specifically structuring interventions for fathers in the prenatal or perinatal period.
General Judgment Comments
The paper is well written; the style is fluent. The design is appropriate, and the procedure is very meticulous and reported in detail. The abstract is explanatory for the study, while the title can be improved. Results are reported following the required standards. The introduction can be enriched with further evidence elucidating the background and the need for the study. The statistical analyses are appropriate and adequately explained. Figures and Tables are informative, although minor edit is required. The discussion section is very interesting and well-developed.
A few minor issues need to be addressed, as listed below.
Minor Issues
- Line 43: when firstly presented, the acronyms should be fully explained (i.e., OECD)
- Lines 64-70: It would be helpful to include more evidence concerning paternal self-efficacy and self-confidence, not only in the Japanese context. Please add more evidence.
- In Table 1, it is unclear what statistical analysis the author adopted (t-test or X2) and how the p-value was computed; please clarify and add the t-test or X2
- In all the Tables, it would be helpful to use the “*” to identify significant results more quickly.
- Lines 246-250: please add a reference for the measures taken from April 2022.
Suggestions
- The TITLE could be slightly edited to provide information on the study’s outcomes.
- To emphasize the study results and their implication in the clinical work, the authors could move the text in lines 245-257 in the “Conclusion.”
Final comments
The author should edit the article following the comments listed above and resubmit the manuscript for further consideration.
Author Response
Point 1: Line 43: when firstly presented, the acronyms should be fully explained (i.e., OECD)
Response 1: As you kindly pointed out, we spelled out the official name as follows.
Line 43: Currently, most Organization for Economic Co-operation and Development (OECD) countries launched legal paternity leave policy to promote gender equality in childcare
Point 2: Lines 64-70: It would be helpful to include more evidence concerning paternal self-efficacy and self-confidence, not only in the Japanese context. Please add more evidence.
Response 2: Thank you for your comments. We added evidence on paternal self-efficacy, citing the study in New Zealand and Finland.
Line 70: Another cross-sectional study in New Zealand found that about 30 to 40% of fathers expressed moderate or low levels of confidence in dealing with children’s behavior and emotions [18]. Fathers in Finland also reported lower parenting self-efficacy than mothers [19].
Point 3: In Table 1, it is unclear what statistical analysis the author adopted (t-test or X2) and how the p-value was computed; please clarify and add the t-test or X2
Response 3: We added the words to clarify which statistical analysis we adopted for Table 1.
Line 149: Descriptive statistics for the group taking paternity leave and the group not taking paternity leave were calculated and compared using the t-test for continuous variables and the chi-square test for categorical variables.
Table1: We added the following caption.
b The p-value was calculated using the t-test for continuous variables and the chi-square test for categorical variables.
Point 4: In all the Tables, it would be helpful to use the “*” to identify significant results more quickly.
Response 4: We added “*” for the significant values. We also added the following caption in Table 2.
Table 2: * The coefficient is significant at the 0.05 level (bilateral).
Point 5: Lines 246-250: please add a reference for the measures taken from April 2022.
Response 5: We added a reference as follows.
- Ministry of Health Labour and Welfare. Outline of the Act on Childcare Leave, Caregiver Leave, and Other Measures for the Welfare of Workers Caring for Children or Other Family Members. Available online: https://www.mhlw.go.jp/bunya/koyoukintou/pamphlet/dl/02_en.pdf (accessed on 24 Mar 2022).
Point 6: The TITLE could be slightly edited to provide information on the study’s outcomes.
Response 6: Thank you for your helpful comment. We edited the title to provide the negative association of paternity leave with father-infant bonding by adding the word “impaired”.
Title: “Association of paternity leave with impaired father-infant bonding: Findings from a nationwide online survey in Japan”
Point 7: To emphasize the study results and their implication in the clinical work, the authors could move the text in lines 245-257 in the “Conclusion.”
Response 7: Thank you for your kind suggestion. We moved the last paragraph in the “Discussion” section to the “Conclusion” section.
- Conclusions
The present study found that fathers who took paternity leave were more likely to report impaired bonding, particularly in the aspects of increased emotion of anger and rejection, with child under two years of age. The current findings have possible social implications for promoting father-infant bonding in the context of rapid increase in the use of paternity leave. From April 2022, various measures will be taken to make it easier for both parents to take childcare leave in Japan, including more flexible framework for taking childcare leave, obligation for employers to confirm the intention of employees to take childcare leave, and disclosure of the status of childcare leave taken [47]. This may lead to an increase in the number of fathers taking childcare leave in the near future. However, the experience of spending a prolonged duration of time with children during paternity leave can still be challenging and demanding for fathers who are usually dedicated to their work. Interventions that mitigate the negative impact of paternity leave would prevent deteriorating father-infant bonding. For example, encouraging fathers before taking paternity leave (such as during maternal hospitalization) to participate in parenting classes may not only help them acquire parenting skills, but also build a social network outside the workplace and prevent loneliness during paternity leave. To make it easier for fathers on leave to attend local childcare salons, it might be advisable to have “father-child only” hours or assign male childcare workers.
Reviewer 3 Report
Thank you for the opportunity to read and learn with this study. This cross-sectional study proposes to verify the father-infant bond between parents and babies under 2 years of age and verifying the relationship with paternity leave. I believe that this article is very interesting as bonding plays a central role in parenting. However, I have some comments and I believe the authors can improve the manuscript.
Introduction is well written and appropriate to the manuscript as point out that providing time to fathers to involve with their born children is helpful for bonding.
Methods: Could the authors please clarify the parents recruitment strategy? How were panelists recruited? When were the three surveys carried out? This aspect should be further explained.
The instrument used was previously validated for parents of four-month-old babies. This study included parents whose youngest child was less than 2 years old. Could these differences in the children's age influence the performance of this questionnaire?
Could the authors explain how long the father can take childcare leave after the baby is born in Japan? Did all respondents have a similar amount of time for parental leave?
Results: Could the authors describe the results or the gradation of the total score on the questionnaire in the groups stratified by the age of the youngest children?
Discussion: it is argued that paternal loneliness influences the father-infant bond. Why does child care leave reduce opportunities to meet friends? I believe that the period of paternity leave can influence the bond. Could any preventive intervention be proposed?
The subject of the study is relevant and little studied. Knowing the characteristics of specific populations contributes to the understanding of social relationships, as cultural and behavioral differences due to work or living habits can influence the father-infant bond.
Author Response
Point 1: Methods: Could the authors please clarify the parents recruitment strategy? How were panelists recruited? When were the three surveys carried out? This aspect should be further explained.
Response 1: Thank you for your comments. We added a little detailed explanation of how panelists were recruited. We also added when the three surveys of the JACSIS study were carried out.
Line 82: Briefly, the JACSIS survey consisted of three surveys in the following target populations: (1) youth and adults aged 15 to 79, (2) women who are currently pregnant or postpartum, and (3) adults living in single-parent households [20], all of which were conducted in 2021. For each survey, the study samples were obtained from the pooled panels of an internet research company (Rakuten Insight, Inc., which holds approximately 2.3 million panelists who voluntarily registered for small incentive points they receive when completing questionnaires).
Point 2: The instrument used was previously validated for parents of four-month-old babies. This study included parents whose youngest child was less than 2 years old. Could these differences in the children's age influence the performance of this questionnaire?
Response 2: Thank you for the comments on the important point. We added some explanations and citations regarding the validity of MIBS-J in fathers with older children.
Line 116: Although MIBS-J was originally developed for mothers of infants between 1 and 4 months of age, a cross-sectional study in Japan showed the same MIBS factor structure in a sample of parents of older children (mean age 3.3 years (standard deviation 2.7 years)) and that it could be used with validity both mothers and fathers and with children of a wide age range [26].
Point 3: Could the authors explain how long the father can take childcare leave after the baby is born in Japan? Did all respondents have a similar amount of time for parental leave?
Response 3: We amended the sentence in the “Introduction” section to explain the system for childcare leave in Japan as follows. Unfortunately, we did not obtain information on the amount of time fathers took for paternity leave in our sample. But we assumed that majority of our samples have taken similar short-term leave, based on Japanese statistics published by the Ministry of Health, Labour and Welfare. We amended the explanation in the Discussion section to clarify this.
Line 46: Japan is no exception; fathers with infants have the right to take childcare leave until the child reaches the age of 1 year (until the day before the child’s first birthday).
Line 257: Third, we did not obtain information on the duration of paternity leave. According to a national survey in 2015, about 80% of all fathers taking childcare leave took only a total of less than two weeks[43]. Given the report that father-infant bonding deteriorates with longer duration of paternity leave [11], our results in a sample in which the majority of participants are assumed to have taken similar short-term leave might underestimate the negative effect of paternity leave on father-infant bonding impairment.
Point 4: Results: Could the authors describe the results or the gradation of the total score on the questionnaire in the groups stratified by the age of the youngest children?
Response 4: Thank you for your helpful comments. We added the mean score and standard deviation of MIBS-J and its subscale in Table 3. We also amended the description in the Results section to explain this result.
Line 185: When child’s age was stratified by 6-months increments, the mean MIBS-J score was 4.7 (< 6 month), 4.1 (6–12 months), 4.6 (12–18 months), and 5.9 (18–24 months), respectively. Although the association became null, the point estimate was similar in each group (β = 0.42 (child age < 6 month), β = 0.58 (6–12 months), β = 0.48 (12–18 months), β = 0.66 (18–24 months), respectively).
Table 3:
|
Table 3. Stratified analysis of the association between paternity leave taking and father-infant bonding by infant age |
||||||||||||||||
|
  |
  |
Infant age |
  |
|||||||||||||
|
< 6 months (n = 339) |
  |
6 – 12 months (n = 418) |
  |
12 – 18 months (n = 321) |
  |
18 – 24 months (n = 116) |
p for trend |
|||||||||
|
Outcome |
Mean (SD) |
β |
95% CI |
  |
Mean (SD) |
β |
95% CI |
  |
Mean (SD) |
β |
95% CI |
  |
Mean (SD) |
β |
95% CI |
|
|
MIBS-J |
4.7 (4.7) |
0.42 |
-0.44 to 1.28 |
4.1 (4.6) |
0.58 |
-0.22 to 1.38 |
4.6 (4.9) |
0.48 |
-0.36 to 1.31 |
5.0 (4.3) |
0.66 |
-0.87 to 2.20 |
0.98 |
|||
|
LA |
1.8 (2.3) |
0.09 |
-0.36 to 0.55 |
1.5 (2.2) |
0.22 |
-0.20 to 0.63 |
1.7 (2.3) |
-0.03 |
-0.48 to 0.41 |
1.7 (2.0) |
0.21 |
-0.53 to 0.94 |
0.33 |
|||
|
AR |
2.3 (2.2) |
0.17 |
-0.27 to 0.61 |
2.0 (2.3) |
0.28 |
-0.13 to 0.68 |
2.4 (2.3) |
0.34 |
-0.09 to 0.77 |
2.7 (2.1) |
0.27 |
-0.52 to 1.06 |
0.19 |
|||
|
MIBS-J, the Japanese version of Mother-Infant bonding scale; LA, lack of affection; AR, anger and rejection; SD, standard deviation. |
||||||||||||||||
Point 5: Discussion: it is argued that paternal loneliness influences the father-infant bond. Why does child care leave reduce opportunities to meet friends? I believe that the period of paternity leave can influence the bond. Could any preventive intervention be proposed?
Response 5: We amended the Discussion section to clarify why fathers on leave have fewer opportunities to meet friends or other fathers to talk to. We also added possible preventive interventions in the Conclusion section.
Line 217: Secondly, paternal loneliness during paternity leave may deteriorate father-infant bonding [31]. It was reported that only about half of fathers have a friend they can talk to about childcare and 60% of their friends are colleagues from work [17]. While a father is on leave, his friends are at work, so he may have less chance to see his friends. Furthermore, fathers taking paternity leave are still a minority in Japan, and there are few opportunities to meet and talk with other fathers on leave while at the park or childcare salons.
Line 292: Interventions that mitigate the negative impact of paternity leave would prevent deteriorating father-infant bonding. For example, encouraging fathers before taking paternity leave (such as during maternal hospitalization) to participate in parenting classes may not only help them acquire parenting skills, but also build a social network outside the workplace and prevent loneliness during paternity leave. To make it easier for fathers on leave to attend local childcare salons, it might be advisable to have “father-child only” hours or assign male childcare workers.
Round 2
Reviewer 3 Report
Thank you for the opportunity to read the revised version of this study.
The authors have adequately addressed the comments and suggestions.
I am happy to recommend acceptance.